# Abdominal Symptoms and Colonic Diverticula in Marfan’s Syndrome: A Clinical and Ultrasonographic Case Control Study

**DOI:** 10.3390/jcm9103141

**Published:** 2020-09-28

**Authors:** Giovanni Maconi, Alessandro Pini, Elia Pasqualone, Sandro Ardizzone, Gabrio Bassotti

**Affiliations:** 1Gastroenterology Unit, Department of Biomedical and Clinical Sciences, ASST Fatebenefratelli-Sacco, University of Milan, 20157 Milan, Italy; pasqualone.elia@gmail.com (E.P.); sandro.ardizzone1@unimi.it (S.A.); 2Cardiovascular-Genetic Center, IRCCS Policlinico San Donato, 20097 San Donato Milanese, Italy; alessandro.pini@grupposandonato.it; 3Gastroenterology & Hepatology Section, Department of Medicine, University of Perugia Medical School, 06122 Perugia, Italy; gabassot@tin.it

**Keywords:** Marfan’s syndrome, diverticula, abdominal symptoms, ultrasound

## Abstract

Background: Marfan’s syndrome (MFS) seems to be frequently associated with colonic diverticulosis, but the prevalence of diverticula and symptoms evocative of diverticular disease in this population are still unknown. Methods: This prospective case control study included 90 consecutive patients with MFS, 90 unselected controls, and 90 asymptomatic subjects. The clinical characteristics, including lower gastrointestinal symptoms, and ultrasonographic features of the bowel, including diverticula and thickening of the muscularis propria of the sigmoid colon, were investigated. In addition, the genotype of MFS patients was assessed. The characteristics of patients and controls were compared using parametric tests. Results: Complaints of abdominal symptoms were made by 23 (25.6%) patients with MFS and 48 (53%) control subjects (*p* < 0.01). Constipation and bloating were reported less frequently by MFS patients than controls (constipation: 13.3% vs. 26.6%, *p* = 0.039; bloating: 3.3% vs. 41.1%, *p* < 0.0001), while other symptoms were not significantly different. Sigmoid diverticulosis was detected in 12 (12.3%) patients with MFS, as well as in 3 (3.3%) asymptomatic healthy subjects and 4 (4.4%) random controls (*p* = 0.0310). The genetic variants of MFS were not correlated with symptoms or diverticula. Conclusion: Patients with MFS have a greater prevalence of diverticula, although less abdominal symptoms, compared to the general population. Symptoms and diverticula in MFS are not correlated with any genetic variant.

## 1. Introduction

Marfan’s syndrome (MFS) is an autosomal dominant hereditary disorder, characterized by an altered production of connective tissue—particularly fibrillin—a major constituent of elastic tissue. Its estimated frequency in the general population is approximately 2–3/10,000 [1].

Patients with MFS may display a wide range of clinical manifestations involving different organs and tissues. For example, major clinical manifestations occur in the cardiovascular (e.g., mitral valve prolapse and aortic root aneurysm and dissection), musculoskeletal, and ocular (e.g., ectopia lentis and myopia) systems. Skin, pulmonary, and neurological systems can also be affected. The phenotype is highly variable, even within affected families.

Marfan syndrome is the result of a mutation in FBN1, the gene encoding the fibrillin-1 protein. Fibrillin-1 belongs to the family of fibrillins, a group of large extracellular proteins that form the core of microfibrils. The latter are ubiquitously distributed, regulate the bioavailability of transforming growth factor beta (TGFB) and bone morphogenic proteins, and provide a scaffold for elastogenesis in the majority of elastic tissues [2,3].

The ubiquitous distribution of the fibrillin-1 protein explains why MFS displays multisystemic involvement. However, it is generally thought that gastrointestinal (GI) tract complications are not commonly associated with MFS. A review of the literature shows instances of diaphragmatic hernia, inguinal hernia, and abdominal wall defects due to the diastasis recti and laparocele of the abdominal rectus muscles. 

Alteration in FBN1 might also cause GI manifestations. In fact, the disrupted connective tissue could potentially result in a structural defect in the GI system and an increased prevalence of diverticulosis in both the colon and the small bowel. Diverticula formation, in turn, might then represent a risk factor for bacterial contamination of the small bowel, intussusceptions, and volvulus [4]. In addition, the abnormalities of the connective tissue could also explain some non-specific GI symptoms in MFS. 

It is worth noting that, at present, the GI manifestations of MFS are poorly documented. Thus far, in the literature, there are only case series reporting the association between MFS, abdominal pain and altered evacuation, and, in particular, cases of severe acute diverticulitis or complications of diverticulosis, for which MFS is historically considered a risk factor [4,5,6,7,8,9,10,11,12]. Inayet et al. [13] and Nee et al. [14] recently evaluated the prevalence of functional GI diseases and pelvic floor symptoms in a cohort of MFS patients compared to patients with Ehlers–Danlos syndrome, a group of inherited heterogenous multisystem disorders often in differential diagnosis with MFS (as far as skin hyperextensibility, atrophic scarring, joint hypermobility, and generalized tissue fragility are concerned). Furthermore, a recent work [15] described the case of a 68-year-old man with MFS and a history of diffuse diverticulosis of the small bowel leading to a perforated distal jejunum.

To date, the prevalence of GI symptoms in MFS, particularly those evocative of diverticular disease, is unknown. Considering that MFS represents a risk factor for diverticulosis even at a young age, and that patients with diverticula are at higher risk of perforation during colonoscopy [16], it would be interesting to develop a noninvasive diagnostic method to better estimate the actual prevalence of morphologic changes in the bowel and the association with GI symptoms in MFS. This would allow optimizing the diagnostic work-up and follow-up on these patients.

Intestinal ultrasound (IUS) has been proposed in recent years as a non-invasive method for the diagnosis of many GI diseases. Of interest, some international guidelines recommend IUS as the first diagnostic technique in patients with suspected acute diverticulitis [17,18]. Moreover, IUS is able to diagnose left-sided colonic diverticulosis in 85% of patients [19] by detecting diverticula as protrusions of the intestinal wall containing fecaliths, combined with an increased thickness of the walls of the sigmoid and left colon, generally due to hypertrophy of muscularis propria [20]. 

Thus far, the data regarding the prevalence of GI symptoms and the morphological features of the colon, including the presence of diverticula of the sigmoid colon in MFS, are scant, as well as their potential correlation with the genetic variability of this syndrome.

In this prospective case–control study, we assessed: (1) The prevalence of lower GI symptoms; (2) the presence of sonographic changes of the bowel wall—including diverticula—in a series of consecutive patients with MFS, compared to a sex- and age-matched sample of the general population; (3) possible correlations between the clinical and ultrasound features of the sigmoid wall and the genotype of MFS.

## 2. Experimental Section

### 2.1. Study Design

From November 2016 to March 2017, we conducted a prospective case–control observational study on a series of consecutive patients with MFS regularly attending the Marfan Clinic at Luigi Sacco Hospital, Milan, which currently includes approximately 350 patients. All patients met the revised Ghent criteria of 2010 for a clinical diagnosis of MFS [21].

The clinical characteristics and ultrasonographic features of the bowels of these patients, including the prevalence of colonic diverticulosis, were compared to those of non-selected subjects matched by sex and age, recruited among students, technicians, nurses, and physicians attending our department and the biochemical laboratory of our hospital. The sonographic features of the sigmoid wall were also compared to those of asymptomatic subjects, identified from the same non-selected population matched for age (±5 years) and gender. The two control groups were conceived to address the following aims: (1) The assessment of prevalence of lower GI symptom had as controls a random population (which may have abdominal symptoms likely due to IBS, diverticular disease and other causes); (2) the assessment of sonographic changes of the bowel wall (diverticula and thickening of the muscularis propria of the sigmoid colon) had as control group asymptomatic subjects (likely healthy and without intestinal diseases).

The study protocol was approved by the local ethics committee (p.n. 19-ST-042) and all patients gave their written informed consent to participate in the study.

### 2.2. Patients

Patients with the following criteria were considered eligible: MFS diagnosed using the Ghent criteria [21]; good clinical conditions and autonomous walking ability (therefore able to perform ambulatory visits); aged between 7 and 70 years; able to provide informed consent or parents’ consent in the case of a minor patient. Patients with major GI surgery were excluded.

### 2.3. Clinical Evaluation

Patients and controls were administered a questionnaire (Appendix A) to investigate previous or current lower GI symptoms and/or diagnoses. The questionnaire included the collection of biographical (i.e., sex age), biometrics (i.e., body mass index (BMI)), and clinical data. The latter specifically assessed: (1) Previous (last year) or current GI symptoms lasting more than one week, including abdominal pain, bloating or abdominal distension, or changes in bowel habit (i.e., diarrhea and/or constipation), severe enough to prompt diagnostic evaluation or treatment; (2) previous colonoscopy, barium enema, or virtual colonoscopy; (3) previous fecal occult blood test for the screening of colorectal cancer; (4) minor GI surgery (e.g., appendectomy, cholecystectomy, and abdominal laparoplasty).

### 2.4. Ultrasound Evaluation

In all recruited subjects, IUS was carried out to detect the morphological characteristics of the small bowel and the colon. All procedures were done during fasting by a sonographer with particular expertise in the bowel (>10,000 investigations) using an ultrasound machine (Hitachi Logos HiVision C, Steinhausen, CH) with a low-frequency convex ultrasound probe (3.5–5 MHz) and a microconvex high-frequency probe (4–8 MHz), as per standard protocol.

The following parameters were retrieved: Dilatation of the small bowel > 3 cm; thickening of any intestinal wall segment > 3 mm; enlarged lymph nodes (shorter in diameter than >7 mm), mesenteric hypertrophy, or peri-intestinal effusion; sigmoid wall thickness; thickness of the muscularis propria of the sigmoid colon; the presence of diverticula and their complications, if any.

### 2.5. Genetic Characteristics

The following genetic variables, assessed in the population at diagnosis and therefore prior to the beginning of the study, were taken into consideration: (1) Positivity of the genetic test; (2) gene mutations; (3) number of mutations; (4) exons; (5) haploinsufficiency/dominant negative mutation. The genomic DNA of each patient was extracted from peripheral blood lymphocytes using a Gene Catcher gDNA 96 × 10 mL Automated Blood Kit (Invitrogen, Life Technologies™, Carlsbad, CA, USA). The genetic test was performed using the next-generation sequencing (NGS) technique. A panel composed of 11 genes (i.e., FBN1, TGFBR1, TGFBR2, COL1A1, COL1A2, COL3A1, COL5A1, COL5A2, MYH11, NOTCH1, and ACTA2) known to be associated with MFS and Marfan-like phenotypes (such as Ehlers–Danlos syndromes (EDS), Loeys–Doetz syndrome (LDS), and thoracic aortic aneurysm (TAAD)) were analyzed. An Illumina TruSeq Custom Amplicon Kit (Illumina, Inc. San Diego, CA, USA) was used to capture all exons, intron–exon boundaries, and at least 50 bp flanking sequences of target genes (RefSeq database, hg19 assembly). The data collected from the NGS experiments were analyzed in order to identify single-nucleotide variants and small insertions/deletions [22].

### 2.6. Statistical Analysis

Data are presented as means ± standard deviation (SD). The clinical and sonographic characteristics of patients and controls were compared with parametric tests (i.e., the *t*-test and the chi-square test). The association between the clinical and sonographic parameters and the genetic variables of patients was assessed by analysis of variance. A value of *p* < 0.05 was considered significant. 

## 3. Results

In the study period, 90 patients with MFS (48 men; aged 36.4 ± 16.4 years; BMI 22.2 ± 4.4) and 180 control subjects, including 90 subjects from a random sample of the general population (48 men; aged 36.55 ± 16.7 years; BMI 22.9 ± 3.4) and 90 asymptomatic healthy subjects (48 men; aged 37.6 ± 16.8 years; BMI 22.6 ± 3.1), were recruited.

### 3.1. Clinical Features

Overall, past or current complaints of abdominal symptoms were made by 23 (25.6%) patients with MFS and 48 (53%) control subjects (*p* < 0.01). In particular, constipation and bloating/abdominal distension were reported less frequently by patients with MFS compared to the controls (constipation: 13.3% vs. 26.6%, *p* = 0.039; bloating/abdominal distension: 3.3% vs. 41.1%, *p* < 0.0001). Meanwhile, diarrhea was reported by 4.4% of patients with MFS and 8.8% of the controls (*p* = 0.371), and abdominal pain by 18.8% of patients with MFS and 21.1% of the controls (*p* = 0.852) (Table 1). Nine patients with MFS and 20 controls had previous endoscopic or radiologic investigations of the colon, mainly for prevention of colorectal cancer (15), rectal bleeding (6), or abdominal complaints (abdominal pain and/or diarrhea) (8). No complications were reported during endoscopic procedures in MFS patients.

### 3.2. Sonographic Features

Overall, sigmoid diverticulosis was detected in 12 (12.3%) patients with MFS, 3 (3.3%) asymptomatic healthy subjects, and four random controls (*p* = 0.0310). The rate of diverticulosis increased with age, starting from 30 years old with greater prevalence over 50 years (Table 2). 

Thickening of the muscularis propria of the sigmoid colon increased progressively with age, but it was not statistically different between patients with MFS and healthy subjects (1.38 ± 0.55 mm vs. 1.42 ± 0.4 mm; *p* = 0.723) or in any age group (Table 3). No other bowel abnormalities were assessed by IUS in the study population, apart from the presence of enlarged mesenteric lymph nodes found in two patients with MFS and four in control subjects. In particular, no significant abnormalities (including bowel wall thickening, abnormal dilatation, or diverticula) were detected in the small bowel.

### 3.3. Genetic Features and Sonographic Findings

Of the subjects, 97.1% were found to harbor a pathogenetic variant in the FBN1 gene. This is in line with the data from the literature, according to which states that not all subjects with a clinical diagnosis of MFS receive a positive genetic FBN1 test. 

In one subject without mutation of the FBN1 gene, a pathogenetic variant of the TGFBR2 gene (a gene involved in transforming the growth factor β (TGF-β) signaling pathway) was present; this led to a diagnosis of LDS, a genetic condition overlapping MSF in an aortic root aneurysm and risk of dissection, in skeletal features and habitus [23]. The suggested follow-up in LDS syndrome is the same as for MFS. In eight patients (8.9%), two genetic variants were identified: A pathogenetic variant in FBN1 and another variant of uncertain significance in the COL1A2, COL5A1, or COL5A2 genes. These genes, when functionally altered, cause Ehlers–Danlos syndrome—the classic type and, on the basis of a recent study [23], could act as modifiers of phenotypes. The presence of diverticula was found in eight out of the 62 (12.9%) patients with a single mutation and in one out of eight (12.5%) patients with a double mutation (*p* > 0.99). 

The thickness of the muscularis propria in patients with a single or double mutation was similar (1.39 ± 0.57 mm vs. 1.38 ± 0.53 mm; *p* = 0.951), and the association between the number of mutations and the thickness of the muscularis propria was not statistically significant (*p* = 0.92). Investigation of haploinsufficiency was available for 65 patients with a mutated FBN1 gene: 20 (33.3%) had a mutually unstable mutation and 45 (66.6%) had a dominant negative mutation. The presence of diverticula was found in two patients (10%) with a mutually tolerant mutation and in seven patients (15.5%) with a dominant negative mutation (*p* = 0.709). The average thickness of the muscularis propria of the sigmoid colon in patients with a mutually tolerant mutation (1.21 ± 0.52 mm) and those with a dominant negative mutation (1.46 ± 0.58 mm) was not statistically different (*p* = 0.091). The type of mutation in the FBN1 gene or the number of genes involved were not correlated with any specific GI symptom.

## 4. Discussion

MFS, together with EDS and LDS, are multisystemic genetic disorders that affect the soft connective tissue, and potentially even the GI tract. However, whereas joint hypermobility syndrome and EDS display a high prevalence of functional GI disorders, in MFS, the prevalence of GI disorders is still disputed [13,14,24,25]. To date, GI disease associated with MFS has been reported only in case series, highlighting, in particular, its association with complications of intestinal diverticula, a condition for which MFS is historically considered a genetic risk factor [4,5,6,7,8,9,10,11,12]. Furthermore, the level of evidence supporting the relationship between MFS and diverticulosis, and its potentially related symptoms, is still unsatisfactory. 

Our study, carried out in a consecutive and unselected series of patients with MFS, showed that despite a higher prevalence of colonic diverticulosis, patients with MFS have a low prevalence of GI symptoms, inferior to that of the general population comparable by gender and age, particularly regarding constipation and bloating. In this regard, we assumed that subjects attending and working at our hospital department were representative of the general population. Despite the fact that this could be questionable, it has to be acknowledged that subjects of this group were recruited merely by opportunity and based on prompt availability criteria and selected only according to their sex and age so as to be comparable to the MFS patients. In particular, none of them were selected according to any symptoms, visits, or examinations previously performed, and all invited subjects agreed to take part in the study. The protocol did not foresee any additional investigation besides ultrasound, and given the features of the symptoms, none of them were submitted to further diagnostic examinations. Therefore, it was not possible to verify whether symptoms were due to diverticulosis.

The prevalence of GI symptoms in cases of MFS has been recently investigated using an electronically mailed questionnaire sent to members of the local and national MFS and EDS societies in the USA. Despite the lack of a clear comparison with a matched control population, the study reported that functional GI and pelvic floor symptoms were significantly higher in Ehlers Danlos Syndrome (EDS) patients than in MFS patients [14]. However, the prevalence of some functional GI disorders complained of by MFS patients, such as constipation (5.3%), diarrhea (1.5%), abdominal pain and overall IBS symptoms (27%), and bloating (16%), were less than half of those of HDS patients, but similar, if not lower, than those usually found in patients referred to a luminal gastroenterology clinic or encountered in the general population [26].

At present, the reasons for the discrepancy between the prevalence of symptoms and functional GI disorders in HDS and MFS patients remains unknown. In addition, the discrepancies in the prevalence of some symptoms (constipation and bloating) found in our study between MFS patients and the controls are difficult to explain. The assessment of symptoms in a series of consecutive patients (and not selected from among those who responded to a survey) and the selection of a control group among personnel attending hospital and their relatives might have influenced the low and slightly high rate of symptoms in the MFS patients and the controls, respectively. 

Interestingly, on account of the low prevalence of GI symptoms, it is unlikely that stress and anxiety, frequently encountered in patients with MFS due to the severe cardiovascular complications of this syndrome [27,28], are predisposing factors of GI functional syndromes such as irritable bowel syndrome in MFS. 

We also investigated the relationship between MFS and diverticulosis, and, in particular, assessed the hypothesis of the syndrome as a genetic predisposition for this condition (verifying the results against asymptomatic subjects) by using IUS. Ultrasound of the GI tract is a well-recognized tool to investigate chronic inflammatory conditions of the gut, including acute diverticulitis, and it is also able to define the anatomical features of the bowel, particularly the thickening of its layers. Given the association between colon diverticulosis and the thickness of the muscularis propria, we assessed the sonographic features of these conditions in a non-selected population of patients with MFS, compared with those of a population of healthy asymptomatic controls. We found that patients with MFS have a high prevalence of colonic diverticulosis, also at younger age, although the associated thickening of the muscularis propria is comparable to that of the general population. 

According to the literature, a fibrillin disorder, which is the cause of MFS, should cause higher levels of TGF-ß. This extracellular mediator interferes with the production of collagen, particularly type I and type III, and this should theoretically be translated into an overall thinning of the organ, particularly the muscularis propria. Our study showed that the thickness of the sigmoid wall of MFS patients is quite similar to that of healthy asymptomatic subjects. The evaluation of any correlation between the clinical and ultrasound characteristics of the sigmoid colon and the patient’s genotype (number of mutations, the lack of a genotype, or a negligent dominance) did not show any statistically significant results, in contrast to those characteristics that most often correlate with a severe clinical (cardiovascular and musculoskeletal) picture.

However, it should be noted that ultrasound, although considered to be a very good diagnostic test, comparable to computed tomography (CT) for acute diverticulitis, is not the same for non-complicated diverticulosis (sensitivity and specificity approximately 85%) [19].

Of course, this study has limitations. First, ultrasound is not the gold standard for diagnosing diverticulosis, and detection can be hampered by gas or patient habitus, as well as by the site and size of diverticula. However, it is the preferred method for the non-invasive investigation of a consecutive and unselected series of patients, including asymptomatic and young subjects. Other techniques such as CT colonography would be more appropriate and accurate, but, given their invasiveness and radiation exposure, their use as screening methods should be justified by clinical conditions. The number of patients we investigated was not large enough for an epidemiological study, but it represents the largest series of patients investigated so far, and we feel that this cohort was sufficient to suggest that other diagnostic investigation to assess the presence of diverticulosis and its complications was not clinically justified, due to the paucity of symptoms in the population and the absence of sonographic signs suggestive of complications of diverticulosis. Third, the IUS assessment of diverticula and intestinal features was not performed blindly with respect to the clinical condition; this was an ineludible condition. On the other hand, a control group was necessary to assess the validity of our findings in the study population. Last, but not least, the assessment of symptoms has been done by using a very simple and not validated questionnaire, not specifically designed to assess diverticular disease, which investigated symptoms (only presence/absence) occurred for more than 1 week in the last year, without considering their duration and severity. 

## 5. Conclusions

In conclusion, the results of this study indicate that MFS is associated with an increased prevalence of colonic diverticulosis, without sonographic signs of diverticulitis or symptoms suggestive of diverticular disease. Of interest, although the population studied is rather young, and therefore with lower risk of symptoms and complications, the patients seem to have less abdominal symptoms and, despite the presence—and likely underestimated prevalence—of diverticulosis, a thickening of the muscularis propria comparable to that of healthy subjects.

## Figures and Tables

**Table 1 jcm-09-03141-t001:** Abdominal symptoms in patients with Marfan’s syndrome (MFS) and control subjects.

	MFS *n* (%)	Random Controls *n* (%)	*p*-Value
Diarrhea	4 (4.4%)	8 (8.8%)	0.371
Constipation	12 (13.3%)	24 (26.6%)	0.039
Abdominal pain	17 (18.8%)	19 (21.1%)	0.852
Bloating	3 (3.3%)	37 (41.1%)	<0.0001

Asymptomatic subjects did not have at present any symptom and did not complain symptoms in the last year.

**Table 2 jcm-09-03141-t002:** Sonographic prevalence of diverticula in patients with MFS and the controls.

Age Class (Number of Patients)	MFS Patients *n* (%)	Asymptomatic Controls *n* (%)	Random Controls *n* (%)
<15 years (*n* = 7)	0 (0)	0 (0)	0 (0)
15–29 years (*n* = 25)	0 (0)	0 (0)	0 (0)
30–44 years (*n* = 21)	2 (9.5)	0 (0)	0 (0)
45–59 years (*n* = 31)	7 (25.9) ^§^	1 (3.2)	2 (7.1)
≥60 years (*n* = 6)	3 (50.0)	2 (33.3)	2 (40.0)
Total (*n* = 90)	12 (13.3) ^#^	3 (3.3)	4 (4.4)

^§^*p* = 0.0310 for asymptomatic controls and random controls; ^#^
*p* = 0.0160 for asymptomatic controls and random controls.

**Table 3 jcm-09-03141-t003:** Sonographic thickening of the muscularis propria in patients with MFS and the asymptomatic controls.

Age Class (Number of Patients)	MFS Patients *n* (%)	Asymptomatic Controls *n* (%)
<15 years (*n* = 7)	0.86 ± 0.33	1.21 ± 0.23
15–29 years (*n* = 25)	1.10 ± 0.40	1.20 ± 0.28
30–44 years (*n* = 21)	1.41 ± 0.47	1.25 ± 0.29
45–59 years (*n* = 31)	1.69 ± 0.54	1.57 ± 0.36
≥60 years (*n* = 6)	1.61 ± 0.44	1.38 ± 0.62
Total (*n* = 90)	1.39 ± 0.55	1.38 ± 0.40

The *p-*value was not significant for any comparison.

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
