# Peer review of "Abdominal Symptoms and Colonic Diverticula in Marfan’s Syndrome: A Clinical and Ultrasonographic Case Control Study"

_jcm, 2020, doi:10.3390/jcm9103141_

Round 1
Reviewer 1 Report
This is a prospective case-control study comparing the prevalence of diverticulosis and possible associated symptoms of Marfan’s syndrome patients with age and sex-matched controls as well as healthy asymptomatic controls. This is an interesting topic with pathophysiologic plausibility given the mutation in fibrillin 1 associated with MFS that could lead to abnormal connective tissue structure thus potentially enhancing defects in the colon wall leading to diverticulosis. Comparing the MFS group to a patient control group as well as healthy volunteers adds strength to the study. Another strength is the use of an expert radiologist to interpret the intestinal US findings.
Suggested Revisions:
- The authors note a significantly higher prevalence of GI symptoms in the control group compared to the MFS group. Please clarify which population was used in the control group. Under Study Design in the Methods section, they state that MFS patients “were compared with those of non-selected subjects matched by sex and age, recruited among students, technicians, nurses and physicians, attending our department and biochemical laboratory of our Hospital. The sonographic features of the sigmoid wall were also compared with those of asymptomatic subjects, identified from the same non-selected population…” Please clarify which patients were used for the control group versus those used for the healthy volunteer group. There may be bias present in the control group leading to increased GI symptoms that are unrelated to the diverticulosis finding.
- The methods state that IUS was carried out to detect the morphological characteristics of the small bowel and colon. The results only report data on the detection of colon diverticula. In the general population, the prevalence of colonic diverticula is higher than small bowel diverticula. How does this compare to the MFS population? Please comment on the results obtained from the small bowel. If there were no small bowel diverticula detected in the study then please state in the results.
Author Response
Dear Reviewer
We thank you very much for the time devoted to our manuscript, for the comments and the constructive suggestions.
Q 1. The authors note a significantly higher prevalence of GI symptoms in the control group compared to the MFS group. Please clarify which population was used in the control group. Under Study Design in the Methods section, they state that MFS patients “were compared with those of non-selected subjects matched by sex and age, recruited among students, technicians, nurses and physicians, attending our department and biochemical laboratory of our Hospital. The sonographic features of the sigmoid wall were also compared with those of asymptomatic subjects, identified from the same non-selected population…” Please clarify which patients were used for the control group versus those used for the healthy volunteer group. There may be bias present in the control group leading to increased GI symptoms that are unrelated to the diverticulosis finding.
R.
The GI symptoms of MFS group have been compared with those of sex and age matched subjects, attending and working in our Hospital Department . The subjects of this group were recruited merely on opportunity and prompt availability criteria, and selected only according their matching by sex and age with MFS patients. In particular none of them has been selected according to any symptoms, visits or exams previously performed, and all invited subjects agreed to take part to the study. Therefore, we felt they could be representative of the general population. The protocol did not foresee any additional investigation besides ultrasound, and given the features of symptoms, none of them has be submitted to further diagnostic exams. Therefore it was not possible to verify whether symptoms were due to diverticulosis.
This has been added in the discussion of the manuscript (page 7; lines 237-246)
Q 2. The methods state that IUS was carried out to detect the morphological characteristics of the small bowel and colon. The results only report data on the detection of colon diverticula. In the general population, the prevalence of colonic diverticula is higher than small bowel diverticula. How does this compare to the MFS population? Please comment on the results obtained from the small bowel. If there were no small bowel diverticula detected in the study then please state in the results.
R. Thanks for this comment and opportunity to clarify.
We assessed by IUS the whole bowel, and all findings have been reported in the results section. We remarked that no significant abnormalities (including bowel wall thickening, abnormal dilatation or diverticula) have been detected in the small bowel (page 5; lines 179-181).
In addition the manuscript has undergone English language editing by MDPI (edited manuscript with revisions attached).

Reviewer 2 Report
interesting study.

Author Response
Reply to the Reviewer #2 (comments added as notes to the manuscript)
1. Comment 1
The incidence of diverticulitis in healthy patients with diverticulosis is not very high. Neither is the risk of perforation during routine colonoscopy. I think it is prudent to cite references and statistics regarding the same. Similar incidence in those in Marfan’s I would imagine should not be significantly higher.
Reply to the comment 1.
We acknowledge that sentence (page 2, line 69-72; “Considering that MFS represents a risk factor for diverticulosis (late diverticulitis and perforation) even in young age, and both diverticula and connective tissue diseases like MFS are at high risk of complication during invasive examinations such as colonoscopy, it would be interesting to develop a noninvasive diagnostic method to better estimate the actual prevalence of morphologic changes in the bowel and the association with GI symptoms in MFS”) is bit forced. Therefore, it has been changed as follows taking into account the referee’s suggestion: “Considering that MFS represents a risk factor for diverticulosis even at a young age, and that patients with diverticula are at higher risk of perforation during colonoscopy [16], it would be interesting to develop a noninvasive diagnostic method to better estimate the actual prevalence of morphologic changes in the bowel and the association with GI symptoms in MFS” (see page 2; lines 71-75)
16. Loffeld RJ, Engel A, Dekkers PE. Incidence and causes of colonoscopic perforations: a single-center case series. Endoscopy. 2011; 43:240-242.
2. Comment 2
Regarding the sentence at page 2 line 80-81 (Of interest, some international guidelines recommend IUS as the first diagnostic technique in patients with suspected acute diverticulitis [16, 17].)“ International GI societal guidelines do not routinely recommend US as the first imaging modality of choice to diagnose acute diverticulitis.
Reply to the comment 2.
We agree that there is not general agreement on this, but some guidelines like those reported in the reference of this manuscript, suggest to use IUS as first imaging technique and CT in indeterminate results. Therefore, we replaced the term “several” with “some”.
3. Comment 3
Study design. What criteria were used to identify the two groups of controls?
Authors need to define random control and asymptomatic control
Reply to the comment 3.
The two control groups were conceived to address the following aims:
1) The assessment of prevalence of lower GI symptom had as controls a random population (which may have abdominal symptoms likely due to IBS, diverticular disease and other causes)
2) The assessment of sonographic changes of the bowel wall (diverticula and thickening of the muscularis propria of the sigmoid colon) had as control group asymptomatic subjects (likely healthy subjects without intestinal diseases).
This has been added in the study design section at page 3 lines 106-112.
4. Comment 4
Regarding the questionnaire. May need a copy of the questionnaire in supplemental files. Drawback would be that this is not a validated tool for assessing symptomatology of diverticular disease
Reply to the comment 4
We thank the reviewer for this comment. The questionnaire used was very simple and included only the assertion of presence or absence of some specific abdominal symptoms.
We acknowledge that this is a major limitation of the study. However, the symptoms considered in this study were the same used in other recent clinical scores developed for the diverticular disease, although not graded for severity and duration. The fact we did not investigate series of symptomatic patients, and we investigated symptoms occurred for more than 1 week (even not consecutive) in the last year, led us to not consider duration and severity of symptoms, because of a likely high recall bias, but in reporting only their absence or presence.
This limitation has been added in the discussion (page 8, lines 319-322) and the (translated) questionnaire has been added as supplementary file.
5. Comment 5
Table.1 The column of asymptomatic controls is missing. What about asymptomatic controls?
Reply to the comment 5.
Asymptomatic subjects have not been included in the table because they did not have at present any symptom and did not complain symptoms in the last year.
This has been added in the footnote of the table.
6. Comment 6
Table.3 The column of random controls is missing. What about random controls?
Reply to the comment 6.
Unfortunately we do not have data about the thickening of the muscularis propria in the random controls.
7. Comment 7
About the sentence: "More importantly, it is unlikely that the abnormal production of connective tissue and the formation of diverticula in the gut, can be related or lead in some way to abdominal symptoms." (Page 7 , lines 254-256).
Diverticulosis of the GI tract may be associated with bacterial overgrowth which can present with similar symptoms.
Reply to the comment 7
We agree with this comment. The sentence has been deleted
8. Comment 8
About the term “TC” in the text (Page 7 , line 277).
What is TC ?
Reply to the comment 8
It was a typos. It was meaning CT (Computed Tomography). This has been corrected
Comment 9
The study is reasonable.
The biggest flaws are the lack of validated tools to 1) assess “GI symptoms” 2) presence of diverticular disease3) guage symptoms of diverticular disease 4) speculate risk of complications.
Several epidemiological studies suggest that the incidence of diverticular disease in the general population is rather high (60-70% ) in >65 years of age wherein only 20% develop symptoms and 15% develop complications. This number is much lower in those <65 years of age.
The numbers from this study suggest the following:
1) a much lower incidence of the same in MFS which begs the clinical utility and application of this pathology in the management of this genetic disorder.
2) This sample was younger. Here the authors compare a disease (with known incidence in older population) in healthy younger subjects(controls).
Reply to the comment 9
The authors thank the reviewer for these comments.
The flaws of the study have been added in the conclusion (pag 8, lines 326-327) including the impact of young age of patients on prevalence of diverticula and correlated symptoms and complications.
